# Influence of Dietary Palm Oils, Palm Kernel Oil and Soybean Oil in Laying Hens on Production Performance, Egg Quality, Serum Biochemicals and Hepatic Expression of Beta-Carotene, Retinol and Alpha-Tocopherol Genes

**DOI:** 10.3390/ani12223156

**Published:** 2022-11-15

**Authors:** Wan Ibrahim Izuddin, Teck Chwen Loh, Henny Akit, Nazri Nayan, Ahmadilfitri Md Noor, Hooi Ling Foo

**Affiliations:** 1Department of Animal Science, Faculty of Agriculture, Universiti Putra Malaysia UPM, Serdang 43400, SGR, Malaysia; 2Institute of Tropical Agriculture and Food Security (ITAFoS), Universiti Putra Malaysia UPM, Serdang 43400, SGR, Malaysia; 3Institute of Bioscience, Universiti Putra Malaysia UPM, Serdang 43400, SGR, Malaysia; 4Sime Darby Plantation Research Sdn Bhd, R&D Centre—Carey Island, Lot 2664, Jalan Pulau Carey, Carey Island 42960, SGR, Malaysia; 5Department of Bioprocess Technology, Faculty of Biotechnology and Biomolecular Science, Universiti Putra Malaysia UPM, Serdang 43400, SGR, Malaysia

**Keywords:** crude palm oil, red palm oil, refined palm oil, palm kernel oil, soybean oil, carotenoids, vitamin A, vitamin E, laying hens

## Abstract

**Simple Summary:**

Limited reports on dietary oil effects in poultry have focused on palmitic acid-, carotenoid- and vitamin E-rich palm oils and medium-chain fatty acid-rich kernel oil. This study explored the effects of dietary crude palm oil (CPO), red palm oil (RPO), refined palm oil (RBD), palm kernel oil (PKO) and soybean oil (SBO) in laying hens. The various oils had no effect on hen-day egg production (HDEP), egg weight, feed intake, feed conversion efficiency (FCR) and egg quality parameters such as shell thickness, shell weight and the percentage of these shell characteristics. Higher egg freshness is indicated by higher albumen height and Haugh unit with dietary CPO inclusion. The RPO and CPO increased the color of the egg yolk as determined by the higher RCF (Roche color fan) and redness (a*) and yellowness (b*) values. Higher serum ALP was observed after feeding the PKO and SBO diets. Dietary CPO and RPO contributed to higher β-carotene in feed, liver and yolk that improved egg yolk color. No difference in retinol and α-tocopherol of serum, liver and yolk was observed. However, a difference was observed in the retinol gene expression in the liver. In conclusion, feeding laying hens with different saturated fatty acids did not influence the production performance and egg quality.

**Abstract:**

Despite being used for many decades, there is a lack of poultry research investigating the effects of dietary palmitic, carotenoid and vitamin E-rich palm oils and medium-chain fatty acid-rich PKO. The current study aimed to elucidate the influence of different dietary oils in layers on production performance, egg quality, serum biochemicals and expression of genes related to β-carotene, retinol and α-tocopherol in the liver. A total of 150 Hisex brown laying hens were fed diets containing CPO, RPO, RBD, PKO or SBO at a similar level for 16 weeks. Different oils did not affect egg production performance and egg quality. CPO improved the freshness of eggs. CPO and RPO enhanced egg yolk color. There was no influence of different oils on serum biochemicals except greater serum ALP in PKO and SBO. CPO and RPO contributed to greater β-carotene in feed, liver and yolk. There was no difference in retinol and α-tocopherol of serum, liver and yolk. However, the liver RBP4A gene was upregulated in CPO and PKO, and the CYP26A1 gene was downregulated in palm oils and PKO. In conclusion, palmitic-rich saturated fatty acids in palm oils and MCFA-rich PKO did not negatively affect egg production performance and quality compared to oil with high unsaturated fatty acids.

## 1. Introduction

Lipid inclusion in the form of oil and fats in poultry feed is essential as an excellent low-cost energy source and provides a wide array of fatty acids [1]. In addition, oil inclusion in the feed has beneficial effects in terms of the physical quality of feed, enhancing palatability, supplying extra caloric effect by improving the absorption of other dietary nutrients and supplying essential fat-soluble vitamins and increasing their absorption [2,3,4]. Hence, adding lipids, especially plant-based oil, to the feed is common in poultry feeding. However, the types of oils used in poultry feed vary, depending on local production and availability of the oil in a particular country. Soybean, rapeseed and linseed oils are commonly used in poultry feed, particularly in temperate countries.

In tropical countries, palm oil is widely used as a source of energy and essential fatty acids in poultry feed. Several types of oils can be produced from the mesocarp (fiber) and the kernel of palm oil fruits. Crude palm oil (CPO) is produced from crude extraction of the mesocarp of palm fruits. Red palm oil (RPO) is a refined red–orange oil produced from the CPO’s molecular distillation process, which retains the carotenoids and phytonutrients. The refining, bleaching and deodorization process of CPO produces refined palm oil (RBD). The benefit of palm oil is mainly the composition of potent antioxidants and balance in fatty acid saturation. Palm oil’s balance in fatty acid saturation has the advantage of less exposure to oxidation concurrently with the presence of antioxidants. It has the richest source of antioxidants with essential fat-soluble compounds such as carotenoids, tocopherols, tocotrienols and water-soluble phenolic compounds. Tocopherols and tocotrienols are isomers of vitamin E, which have high-antioxidant potential and are the primary antioxidants for lipids to protect lipids from oxidation [5]. The carotenoids are part of the antioxidant defense system and efficient antioxidants in scavenging singlet molecular oxygen and peroxyl radicals [6].

On the other hand, palm kernel oil (PKO) is an oil extracted from the kernel of palm fruits which is a highly saturated oil rich in medium-chain fatty acids (MCFA). The MCFA has been shown to reduce the gut pathogen load and contribute to broiler chickens’ gut health and performance [7,8]. However, adding PKO to the poultry diet is less common than palm oils, mainly due to the abundance and cheaper price of palm oils as well as that PKO is mainly used for human food application. Whereas, extraction of oil from soybeans and the refining, bleaching and deodorization process produce soybean oil (SBO) which is rich in polyunsaturated fatty acids (PUFA), mainly in the form of linoleic acid. Figure 1 summarizes the similarity and differences in the properties of the oil of interest.

The published research on dietary palm oil in laying hens was limited and focused on different inclusion levels of the oil or using a single type of palm oil and comparing it with other vegetable oils [9,10,11,12]. Furthermore, a lack of studies compared palm oil with medium-chain-rich kernel oil and highly unsaturated oil in poultry, particularly laying hens. The parameters measured in previous studies were not extensive or in-depth and lacked information on the measurement of the serum biochemical, deposition and metabolism of beta-carotene, retinol and vitamin E, as well as its gene expression. Hence, the current study explored the effects of dietary palm oils, PKO and SBO, on egg production performance, egg quality, serum biochemical, metabolism of β-carotene, retinol and α-tocopherol and meat physicochemical attributes in laying hens.

## 2. Materials and Methods

### 2.1. Ethical Approval, Animals and Management

The use of laying hens in the study was approved by the Institutional Animal Care and Use Committee of Universiti Putra Malaysia (AUP No: UPM/IACUC/AUP-R013/2020). The feeding trial was conducted at the Poultry Unit, Farm 15, Department of Animal Science, Faculty of Agriculture, Universiti Putra Malaysia. A total of 150 Hisex Brown laying hens at 16 weeks of age were purchased from a local commercial layer farm (QL Poultry Farms, Rawang, Selangor). The hens were randomized into five treatment groups (30 hens per group), which contained six biological replicates per treatment and five hens per biological replicate. The hens were kept in an individual cage (30 cm width, 50 cm depth and 40 cm height) of a two-tier A-type battery cage in an open-sided hen house system. The lighting was programmed automatically and consisted of 16 h light and 8 h dark which includes average daylight of ±12 h and an additional 4 h using LED lighting. During the feeding trial, the temperature ranged between 24 to 32 °C and the mean humidity was 80 ± 5%.

### 2.2. Dietary Treatments

Five dietary treatments contained different oil types, namely CPO, RPO, RBD, PKO and SBO, as shown in Table 1. The isocaloric and isonitrogenous diet was formulated using feed formulation software (FeedLIVE, Nonthaburi, Thailand) containing 3% oil from different sources. The nutrient requirement of Hisex Brown laying hens fed 120 g feed/hen daily was met according to the management guide. The experimental diets were prepared monthly and kept in an airtight plastic container at room temperature. The feed was offered daily in the morning (0700–0800) in the mash form, and water was offered ad libitum via a nipple drinker. The adaptation period lasted 6 weeks (week 16 to week 21), and hens received a diet with respective oils. The feeding trial period (data and sample collection) lasted 16 weeks ranging from week 22 to week 37. The experimental design and feeding trial timeline are simplified in Figure 2.

### 2.3. Field Data Collection

Eggs were collected daily in the morning (0900) from the individual hen for egg number and egg weight. Feed intake was determined weekly by collecting feed refusal at the end of the week. The data were used to calculate hen-day egg production (HDEP), egg weight, egg mass, feed intake and feed conversion ratio (FCR). Two eggs from each replicate (a total of 12 eggs per treatment) were collected fortnightly for egg quality analysis. Feed samples were collected during feed preparation at the feed mill and kept in a −80 °C freezer until analysis.

### 2.4. Egg Quality

Egg quality was determined using EggAnalyzer^®^ (ORKA Food Technology, Herzliya, Israel) to yield egg weight, albumen thickness, Haugh unit and yolk color parameters. The yolk color was further measured using calibrated ColorFlex EZ spectrophotometer (HunterLab, Reston, VA, USA) with an optical grade sample cup. The individual yolk sample was read in triplicate per sample with 90° sample rotation for each reading. The color measurement for egg yolk samples was expressed as L* (lightness), a* (redness) and b* (yellowness). Eggshells were dried in a 60 °C oven for 24 h to determine the eggshell weight, percentage of eggshell to overall egg weight and eggshell thickness using a digital vernier caliper (Mitutoyo Digimatic Caliper Series 500, Kawasaki, Japan). Egg yolk samples from week 16 were collected, freeze-dried (Labconco, Kansas City, MO, USA) and kept in an 80 °C freezer for subsequent analysis.

### 2.5. Sacrifice and Sample Collection

At the end of the feeding trial (week 37), one hen from each replicate (6 birds per treatment) was randomly selected and transported to the abattoir of the Department of Animal Science, Faculty of Agriculture, Universiti Putra Malaysia, for sacrificing and collecting samples. The hens were euthanized through the Halal slaughter method, and approximately 8 mL of blood was collected into a 10 mL blood tube (BD Vacutainer^®^ Serum Tubes) at the bleeding point. Blood was kept on ice to clot, centrifuged at 3000× *g* to collect serum, and stored at −80 °C freezer until subsequent analysis. The internal organs were eviscerated. A portion of the lower right lobe of the liver sample was collected, kept in a cryotube and frozen in liquid nitrogen immediately before keeping at −80 °C.

### 2.6. Serum Biochemistry

The serum biochemistry analysis was conducted at the Veterinary Haematology and Clinical Biochemistry Laboratory, Faculty of Veterinary Medicine, Universiti Putra Malaysia, using respective kits on Hitachi 902 Automatic Analyzer (Roche Diagnostics, Basel, Switzerland). Serum samples were analyzed for liver enzymes, namely alkaline phosphatase (ALP), alanine transaminase (ALT), aspartate transaminase (AST) and gamma-glutamyl transferase (GGT). In addition, the total protein, albumin (A), globulin (G), A:G, calcium (Ca), phosphorus (P) and Ca:P were also determined.

### 2.7. β-Carotene Determination

The solvent extraction was conducted using acetone, and the measurement of β-carotene concentration was measured using UV/VIS spectrometric detection according to the protocol described by Biswas et al. [13]. For the extraction procedure, 1 g of samples were weighed in a glass test tube with a screw cap. Chilled acetone was added and held for 15 min with occasional shaking in a chiller at 4 °C and mixed by vortex for 10 min. Next, the tubes were centrifuged at 1370× *g* for 10 min at 4 °C. The supernatant was transferred into a separate test tube; the second extraction was conducted with another 5 mL of chilled acetone, mixed by vortex before centrifugation. The supernatant was combined in a tube and filtered using Whatman filter paper No. 42. The β-carotene (Sigma-Aldrich, St. Louis, MO, USA) standards were prepared ranging from 0.1 to 100 μg/mL in acetone. The absorbance of the extract, blank and standards was measured at 449 nm using a Multiskan™ Go spectrophotometer (Thermo Scientific, Waltham, MA, USA). The concentration of β-carotene in the samples was extrapolated using a standard curve of absorbance vs. the known concentration of β-carotene.

### 2.8. Retinol and α-Tocopherol Concentration by HPLC

The sample preparation and extraction were as described by Grebenstein and Frank [14]. Briefly, 200 mg of liver or yolk sample was transferred in a glass tube with a PTFE-lined screw-cap (on ice) and added with 2 mL of ethanolic 1% (*w*/*v*) ascorbic acid, 900 μL distilled water and 300 μL saturated potassium hydroxide. The tubes were incubated for 30 min in a 70 °C shaking water bath for saponification before being cooled on ice. About 1 mL of distilled water, 25 μL ethanolic 0.001% (*w*/*v*) BHT (Sigma-Aldrich, St. Louis, MO, USA), 300 μL glacial acetic acid (Merck, Boston, MA, USA) and 2 mL n-hexane (Sigma-Aldrich, St. Louis, MO, USA) was added into the tubes and mixed by hand inversion. Serum samples were not saponified, in which 200 μL of serum was transferred in a glass tube and mixed with 2 mL of ethanolic 1% (*w*/*v*) ascorbic acid, 900 μL distilled water and 2 mL n-hexane and mixed by hand inversion. Liver, yolk and serum samples were centrifuged for phase separation at 1500 rpm for 3 min, and a 1.5 mL n-hexane layer was aliquoted into a new glass test tube. A similar extraction step was repeated by adding 2 mL n-hexane into the sample tube and collecting a 1.5 mL n-hexane layer into the glass test tube. The n-hexane was dried under a vacuum using a centrifugal evaporator. The viscous remaining was resuspended in 100 μL of mobile phase and kept in a 2 mL amber glass vial with a PTFE-lined screw cap.

The liquid chromatography was conducted on Agilent 1100 Series HPLC System. The mobile phase was a mixture of HPLC grade methanol and ethanol (Sigma-Aldrich, St. Louis, MO, USA) at a ratio of 75:25, respectively, and running at a constant flow rate of 0.8 mL/min. The separation of peaks was conducted on a Synergy™ 4u Hydro-RP 80A column, 150 mm × 4.6 mm × 4 μm (Phenomenex, Torrance, CA, USA), fitted with a guard column heated at 40 °C. Retinol and α-tocopherol were detected at 325 nm and 292 nm, respectively, using a fluorescence detector (FLD). In addition, a standard curve of retinol and α-tocopherol (Sigma Aldrich, St. Louis, MO, USA) was constructed to extrapolate the concentration of retinol and α-tocopherol in the samples.

### 2.9. Liver mRNA Expression of β-Carotene, Retinol and Tocopherol

Extraction of the total RNA was conducted using NucleoSpin^®^ RNA plus kit (Machery Nagel, Dueren, Germany) according to the protocol outlined by the manufacturer. The extraction kit contained NucleoSpin^®^ gDNA Removal Column for removing DNA contamination and a NucleoSpin^®^ RNA Plus Column to bind and purify RNA before elution with TE buffer (pH 7.5). The total RNA quantity and quality were determined using a Multiskan™ Go spectrophotometer (Thermo Scientific, Waltham, MA, USA). The total RNA (1000 ng) was transcribed to cDNA using the cDNA Synthesis Kit (Biotechrabbit, Berlin, Germany) following the manufacturer-supplied protocol. The qPCR was conducted on LightCycler^®^ 480 Instrument (Roche, Basel, Switzerland) with a 96-well plate format using 4× CAPITAL^TM^ qPCR Green Master Mix (Biotechrabbit, Berlin, Germany). The qPCR reaction mix contained 5 μL CAPITAL qPCR Green Mix, 2 μL forward and reversed primers, 1 μL cDNA and 12 μL nuclease-free water. The cycling program was set at 95 °C for 2 min and 30 s for initial activation. Then, 45 cycles of quantification step comprising of denaturation at 95 °C for 15 s and combined annealing, and extension for 30 s at a temperature specific to the primer used. A melting curve was then conducted to confirm the specificity of the amplification following the instrument’s melt curve program. The information on the primers of housekeeping and target genes is provided in Table 2. The gene expression result is expressed as a fold-change of the treated groups to the control group using the 2^−∆∆Ct^ method [15].

### 2.10. Experimental Design and Statistical Analysis

The experiment was designed for a completely randomized design (CRD). All statistical analysis was conducted on the SAS software package, version 9.4 (SAS Inst. Inc., Cary, NC, USA). The data obtained were checked for distribution using PROC UNIVARIATE and determined based on Shapiro–Wilk. All data were normally distributed. The data were analyzed using one-way analysis of variance (ANOVA) using the General Linear Model (GLM). In addition, Duncan’s multiple range test was performed for treatment means comparisons. The difference was considered significant at *p* < 0.05.

## 3. Results

### 3.1. Production Performance

Various dietary oils did not affect HDEP, egg weight, egg mass, feed intake and FCR (Table 3). The egg weight differed between oils at 30 to 33 weeks of age and for egg mass at 22 to 25 and 26 to 29 weeks of age. At 30 to 33 weeks of age, the notable difference was a lower egg weight that was observed in RPO compared to SBO. A higher egg mass was recorded in RBD than in other oils, but there was no difference in PKO at 22 to 25 weeks of age. At 26 to 29 weeks of age, there was a higher egg mass in RBD and PKO as compared to RPO and SBO but there was no difference in CPO. The overall egg mass was more remarkable in RBD than in RPO and SBO.

### 3.2. Egg Quality

A significant difference between sources of oils was shown in albumen height, Haugh unit and yolk color scale based on RCF and a* and b* based on CIELAB color space (Table 4). The albumen height was higher in CPO than in other oils, other than PKO. No difference was observed in albumen height between SBO to RPO, RBD and PKO. A similar trend was seen for albumen height was shown for the Haugh unit. In terms of yolk color based on RCF, CPO and RPO had greater values, followed by PKO and RBD, and the lowest RCF value was SBO. RPO displayed the highest a* as compared to other treatments. There were no differences in a* between CPO, RBD, PKO and SBO. The b* was highest in RPO compared to other treatments and lowest in CPO and RBD.

### 3.3. Serum Biochemical

The inclusion of different oils had no effects on most of the serum biochemicals except for the serum ALP (Table 5). The PKO showed the highest ALP enzyme level compared to other oils with no difference in SBO. No difference was seen in serum ALP enzyme between CPO, RPO and RBD and lower ALP enzyme level in CPO and RBD compared to SBO.

### 3.4. Vitamin A and β-Carotene Concentration

Different oils did not affect serum, liver and yolk retinol content (Table 6). Feed, liver and yolk had a significant effect on β-carotene content and feed had a significant effect on α-tocopherol. In feed, higher β-carotene was recorded in RPO and CPO and lower in RBD, PKO and SBO. The β-carotene in the liver had a similar trend to the feed. In the yolk, CPO, RPO and PKO had greater β-carotene than RBD and PKO.

### 3.5. α-Tocopherol Concentration

The feed content of α-tocopherol was higher in CPO than in PKO and SBO, but there was no difference between CPO to RPO and RBD (Table 7). No difference was observed in α-tocopherol between RPO, RBD, PKO and SBO. The different levels of α-tocopherol in the feed did not affect the α-tocopherol content in the serum, liver and yolk.

### 3.6. Liver Retinol, β-Carotene and Tocopherol Gene Expression

No difference was observed between oils in BCO1 and TTPA expression, but differences in RBP4A and CYP26A1 were observed (Table 8). There was a higher RBP4A expression in CPO and PKO as compared to RPO and RBD. CYP26A1 had lower fold changes in palm oils and PKO compared to SBO. There was no difference in fold changes of the CYP26A1 gene between CPO, PKO, CPO and RPO.

## 4. Discussion

### 4.1. Production Performance

The similarity in production performance in laying hens was expected despite the difference in the saturation profile of oils in the diet. The current study discovered different palm oils had no notable effects on HDEP, feed intake and FCR. Therefore, no impact on feed conversion could be attributed to the persistence of egg production, egg weight and feed intake between the different dietary oils. Similar findings in comparing RBD and SBO by Agboola et al. [16] showed no difference in egg production, feed intake and FCR in 34-week-old laying hens fed 1.5% RBD and 1.5% SBO. To our knowledge, no previous report was found on the RPO effects in broiler and layer chickens but there was one for laying ducks. The ducks were fed either 3% SBO, 1%, 2% or 3% RPO and with no difference in egg production rate between similar inclusion levels of RPO and SBO. However, the egg production in 3% RPO was higher than 1 and 2% RPO [17]. In addition, there was no difference in FCR between the treatment groups, but daily feed intake was higher in 1, 2 and 3% RPO (with no difference from each other) as compared to 3% SBO [17]. In a previous study, the supplementation of carotenoids and vitamin E did not contribute to the improvement of growth performance in poultry. Karadas et al. [18] reported a lack of effects in growth performance parameters such as body weight, feed intake and FCR in broiler chickens with dietary vitamin E and carotenoids. Hence, current findings suggested that different dietary oils containing different content of fatty acid saturation, carotenoids and vitamin E had no effects on egg production performance, feed intake and FCR.

Nevertheless, the impact of different oils can be seen in egg weight and mass. Greater egg weight in SBO than in RPO was observed at 30–33 weeks of age despite no difference in egg weight for the overall period. There was a higher egg mass in RBD and PKO at 22 to 29 weeks of age and in RBD for the overall period. In contrast to our findings, Yifei et al. [17] found no difference in egg weight between 3% RPO and SBO in laying ducks. Kang et al. [19] also reported no difference in egg weight between laying hens fed 1.5 and 3.5% CPO compared to 2% high-PUFA fish oil despite feeding with different levels of oil in the diet. Despite not agreeing with previous studies, the current study suggests that a higher PUFA fraction of SBO and a higher MCFA fraction could improve egg weight. The egg mass is based on the interaction between HDEP and egg weight. Despite no difference in HDEP, the RBD and PKO showed higher HDEP values and, in combination with egg weight, resulted in a higher egg mass value.

### 4.2. Egg Quality

All treatment groups were similar in overall egg grade, which fell under grade A (between 60.0–71.9). The Haugh unit is based on albumen height and egg weight, indicating the quality and freshness of eggs. The CPO had the highest albumen height and Haugh unit compared to RPO, RBD and SBO. Similar albumen height and Haugh unit were observed in PKO compared to other oils. Contrary to current findings, Agboola et al. (2016) reported no difference in albumen height and Haugh unit of eggs in 34-week-old laying hens fed 1.5% palm oil and 1.5% SBO.

Similarly, in laying ducks, Yifei et al. [17] also reported no difference in albumen height and Haugh unit of eggs between RPO (1, 2 and 3%) and 3% SBO. Our study found no effects of different oils on shell thickness, weight and percentage. Therefore, a similar content of Ca and P in the diet and serum could be associated with a lack of effects on the eggshell parameters. It reflects the similarity in Ca absorption, metabolism and deposition of calcium carbonate to the eggshell [20]. Current findings were in line with Agboola et al. [16], who reported no difference in shell weight and thickness between 1.5% palm oil and SBO in 34 weeks of laying hens.

A higher yolk color (RCF) was observed in CPO and RPO, followed by RBD and PKO and the lowest in SBO. The RPO showed a higher a* than other oils. The b* was also the highest in RPO but lowest in CPO and RBD. Our findings corroborated with previously reported studies. Yeasmin et al. [21] reported the increase in the level of CPO from 1.5, 3 to 5% markedly increased the egg yolk color index. A similar result was reported by Kang et al. [19], indicating a higher yolk color index in laying hens fed 1.5 % and 3.5% CPO than in 2% fish oil. In laying ducks, Yifei et al. [17] also found that the yolk color score was higher in the RPO groups (1–3%) than in the SBO group. In addition, a greater color score was observed in 3% RPO than in 1% RPO. However, Agboola et al. [16] reported no difference in yolk color between laying hens at 34 weeks fed 1.5% palm oil and 1.5% SBO over a 6-week feeding trial period. The carotenoids in the oil contributed to the dietary carotenoids and were deposited into the egg yolk, causing a darker yellow–orange color. The CPO and RPO contained higher content of carotenoids, mainly in the form of α and β-carotene [22], which contributed to the darker color value of the yolk. Lower carotenoids in the refined oils resulted in lesser color pigmentation in the egg yolk.

### 4.3. Serum Biochemical

Serum ALT, AST and GGT did not differ between different oils except for serum ALP, which was higher in PKO and SBO than palm oils. Serum ALP measurement is routinely measured to reflect the ALP activity in the liver, bone and kidney in which a higher level than the normal range indicates disease. The serum ALP level in the current study ranged from 339–736, which fell within the normal range of serum ALP in laying hens with adequate nutrients [23]. Elevated ALP levels of PKO and SBO could be associated with oxidative stress of related organs such as the liver and intestine, as PKO and SBO had lower antioxidant compounds and the presence of higher peroxidation products. Different properties of dietary oils did not influence serum protein components such as TP, albumin, globulin and A:G. However, Kolani et al. [9] discovered that different levels of dietary palm oil in laying hens increased the serum total protein when fed at 3% compared to 1 and 2% of inclusion and no oil inclusion. Hence, the lack of effects on serum protein components could be attributed to similar oil inclusion levels in all treatments. Serum levels of Ca, P and their ratio were not affected by different dietary oils, suggesting no influence on the uptake of dietary Ca and P from the intestinal tract. However, Saminathan et al. [24] indicated that in laying hens, the CPO disadvantage absorption of Ca and P because of saponification with feed minerals. However, our findings suggested otherwise; the saturation profiles of the dietary oils did not influence the absorption and deposition of Ca and P. It was further confirmed by no difference in the shell thickness, weight and percentage between oils.

### 4.4. Retinol, α-Tocopherol and β-Carotene of Feed, Serum, Liver and Yolk

The impacts of different oils on the β-carotene concentration can be seen in the feed, liver and yolk but not in the serum. The CPO and RPO were shown to enhance β-carotene content in feed, liver and yolk. The PKO was shown to increase the deposition of β-carotene in the yolk, similar to CPO and RPO. A previous study showed that the β-carotene content of feed increased in 1, 2 to 3% RPO compared to 3% SBO [17]. Similarly, Kang et al. [19] reported a higher concentration of α- and β-carotene in the feed containing 1.5 % and 3.5% CPO compared to 2% fish oil. Yifei et al. [17] also reported that the egg yolk β-carotene was higher in 2% and 3% RPO compared to 1% PKO and 3% SBO in laying ducks. The β-carotene is a pro-vitamin A carotenoid. It is a precursor for converting β-carotene to retinol. The β-carotene is absorbed and converted into vitamin A in the form of retinoid by intestinal enterocytes and transported to the liver for storage by chylomicrons through the lymphatic system [25]. In the current study, the level of retinol in the serum, liver and yolk was not different between dietary oils despite higher β-carotene levels in the feed and liver as a precursor. However, Kang et al. [19] found a higher concentration of retinol in the yolk of hens fed 1.5 % and 3.5% CPO compared to 2% fish oil but no difference between palm oil inclusion levels. The dose of β-carotene may influence the conversion of β-carotene into vitamin A [26]. In the current study, the lack of effects in serum, liver and yolk retinol despite higher β-carotene in the feed can be attributed to the adequacy of metabolic production of retinol coming from dietary vitamin A in the form of retinyl esters in the diet. In addition, the synthesis of β-carotene to retinol is based on the body’s requirement, and excessive intake was shown not to be detrimental [27]. Excessive intakes of β-carotene do not appear to develop any retinol toxicity effects compared to the dietary retinyl esters supplementation [28].

The presence of α-tocopherol was higher in feed containing CPO and less in PKO and SBO. The feed containing RPO and RBD was not different to other oils. Kang et al. [19] suggested that the amount of vitamin E in the liver and its deposition in egg yolk depended on the dietary vitamin E. Nevertheless, the current study disclosed that the different levels of dietary α-tocopherol did not affect the α-tocopherol level in serum, liver and yolk. Conversely, Kang et al. [19] found a lower concentration of α-tocopherol in the yolk of laying hens fed 1.5 and 3.5% CPO compared to 2% fish oil but with no difference between palm oil inclusion levels. The similar level of α-tocopherol in serum and liver could be interpreted as a similar rate of intestinal absorption and liver synthesis of α-tocopherol. The liver is the central organ of the regulation of lipoprotein uptake, synthesis and secretion and the xenobiotic detoxification system that determine vitamin E, including α-tocopherol level [29]. Therefore, a similar synthesis of α-tocopherol by the liver and secretion into the blood would contribute to a similar deposition rate of α-tocopherol in the yolk.

### 4.5. Liver Retinol, β-Carotene and α-Tocopherol Gene Expression

To our knowledge, there are no studies detailing the effects of dietary palm oil on liver gene expression related to β-carotene, retinol and α-tocopherol in laying hens. The α-tocopherol transfer protein encoded by the TTPA gene is a transport protein that aids the α-tocopherol secretion from hepatocytes into blood circulation through nascent VLDL to maintain the plasma concentrations of α-tocopherol [29]. Our results indicated that different oils did not affect the liver TTPA gene regulation. The lack of significance of liver TTPA gene expression was in line with the no differences in the concentrations of α-tocopherol in the serum, liver and yolk between different oils. BCO1 protein is a crucial cleavage enzyme involved in the first step of cleaving β-carotene into retinal before the conversion of retinal to retinol by retinol dehydrogenase enzyme [27]. Different dietary oils did not affect the regulation of the liver BCO1 gene. However, the concentration of β-carotene in the feed and liver was higher in palm oils which could act as the target of the cleaving action of the BCO1 protein. In the current study, a similar concentration of serum, liver and yolk retinol agreed with the no difference in the regulation of the liver BCO1 gene despite the higher liver β-carotene in CPO and RPO.

Retinol-binding protein is a specific vitamin A transport protein produced mainly in the liver and secreted into chicken blood plasma [30]. The liver RBP4A gene was upregulated in CPO and PKO relative to SBO. Despite the upregulation in the RBP4A gene, the current study disclosed that serum retinol concentrations were similar between the treatment groups. The CYP26A1 protein involves vitamin A catabolism which oxidizes retinoic acid into its inactive forms for excretion through the digestive system [31,32]. Liver CYP26A1 were downregulated in palm oils and PKO relative to SBO. However, downregulation of the CYP26A1 gene did not contribute to the increment in retinol concentrations in the liver, serum and yolk of palm oils and PKO. Although there was a difference in the regulation of genes related to retinol, the concentrations of retinol in the serum, liver and yolk were not different between oils. Therefore, the body’s homeostatic regulation in the production and disposal of vitamin A to maintain the concentrations of vitamin A might relate to the lack of difference in retinol. In addition, the excessive intake of α-tocopherol could contribute to the inhibition of retinol-binding protein synthesis but not on CYP26A1. Zhou et al. [33] reported that dietary 320 and 1280 IU/kg of α-tocopherol in vivo and 100 uM of α-tocopherol in hepatocytes in vitro inhibited retinol-binding protein synthesis. Still, α-tocopherol supplementation did not affect the regulation of the CYP26A1 gene in the liver and hepatocytes.

## 5. Conclusions

Overall, despite the difference in saturation profile of various dietary oils failed to influence the egg production performance of the laying hens. The RBD contributed to the improvement of the egg mass. The CPO improved the freshness of eggs through an increase in HU and AH. Serum biochemicals were not affected, except for higher ALP in PKO and SBO. The presence of natural carotenoids in CPO and RPO improved the presence of carotenoids in feed and deposition in the liver and egg yolk thus enhancing egg yolk color. No effects were seen on serum, liver and yolk retinol and α-tocopherol. However, the liver RBP4A gene was upregulated in CPO and PKO, and the CYP26A1 gene was downregulated in palm oils and PKO groups. Therefore, it can be concluded that palmitic-rich saturated fatty acids in palm oils and MCFA-rich PKO did not influence egg production performance and quality.

## Figures and Tables

**Figure 1 animals-12-03156-f001:**
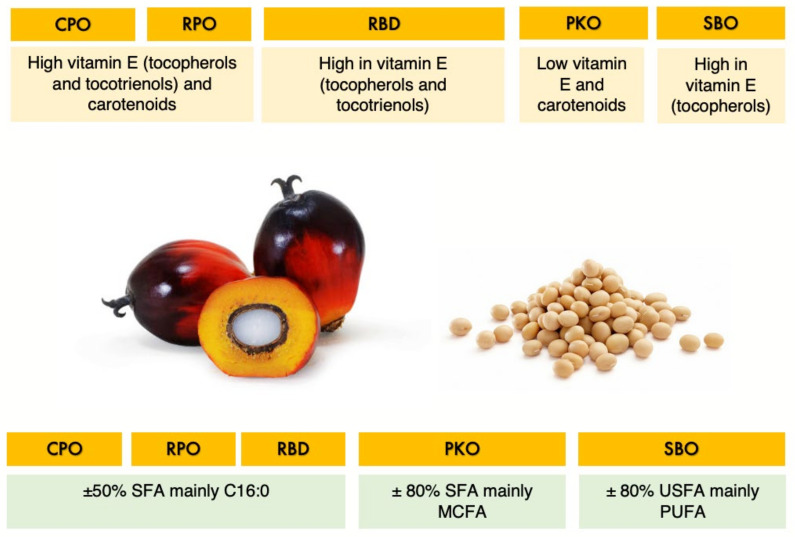
Summary of the properties of oil of interest.

**Figure 2 animals-12-03156-f002:**
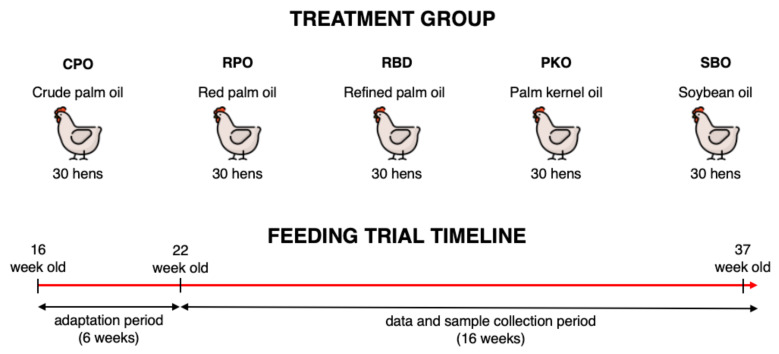
Experimental design and feeding trial timeline.

**Table 1 animals-12-03156-t001:** Ingredients and nutrient profiles of feeds containing different oils.

Treatment	CPO	RPO	RBD	PKO	SBO
Ingredients (%)					
Corn	48.90	48.90	48.90	48.90	48.90
Soybean meal	28.00	28.00	28.00	28.00	28.00
Wheat pollard	8.000	8.000	8.000	8.000	8.000
CPO	3.000	-	-	-	-
RPO	-	3.000	-	-	-
RBD	-	-	3.000	-	-
PKO	-	-	-	3.000	-
SBO	-	-	-	-	3.000
DL-Methionine	0.300	0.300	0.300	0.300	0.300
MDCP	2.300	2.300	2.300	2.300	2.300
Calcium carbonate	8.350	8.350	8.350	8.350	8.350
Choline chloride	0.200	0.200	0.200	0.200	0.200
Salt	0.350	0.350	0.350	0.350	0.350
Mineral mix	0.200	0.200	0.200	0.200	0.200
Vitamin mix	0.200	0.200	0.200	0.200	0.200
Antioxidants	0.100	0.100	0.100	0.100	0.100
Toxin binder	0.100	0.100	0.100	0.100	0.100
TOTAL	100.0	100.0	100.0	100.0	100.0
**Calculated nutrients (in % unless stated)**					
ME (kcal/kg)	2790	2790	2790	2790	2790
CP	17.17	17.17	17.17	17.17	17.17
EE	4.980	4.98	4.98	4.98	4.98
CF	3.800	3.80	3.80	3.80	3.80
Ca	4.000	4.00	4.00	4.00	4.00
Total phosphorus	0.840	0.84	0.84	0.84	0.84
Avail. phosphorus	0.460	0.46	0.46	0.46	0.46
Methionine	0.581	0.581	0.581	0.581	0.581
Lysine	0.933	0.933	0.933	0.933	0.933

CPO: crude palm oil, RPO: red palm oil, RBD: refined palm oil, PKO: palm kernel oil, SBO: soybean oil, MDCP: mono dicalcium phosphate, ME: metabolizable energy, CP: crude protein, EE: ether extract, CF: crude fiber, Ca: calcium. Vitamin mix contains 50,000 mIU vitamin A, 10,000 mIU Vitamin D3, 75 g/kg vitamin E, 20 g/kg vitamin K3, 10 g/kg vitamin B1, 30 g/kg vitamin B2, 20 g/kg vitamin B6, 0.1 g/kg vitamin B12, 60 g/kg calcium d-pantothenate, 200 g/kg nicotinic acid, 5 g/kg folic acid, 235 mg/kg biotin, 25,000 FTU phytase (based on per kg vitamin mix). Mineral mix contains 0.2 g/kg selenium, 80 g/kg iron, 100 g/kg manganese, 80 g/kg zinc, 15 g/kg copper, 4 g/kg potassium, 1.5 g/kg sodium, 1 g/kg iodine and 0.25 g/kg cobalt (based on per kg mineral mix).

**Table 2 animals-12-03156-t002:** The forward and reverse of primer sequence, product size, accession number of target genes.

Target Gene	Primer Sequence	Product Size (bp)	Accession No.
GAPDH F	CTGGCAAAGTCCAAGTGGTG	275	NM_204305.1
GAPDH R	AGCACCACCCTTCAGATGAG		
TTPA F	TCCAGCAGTGGCCAAGAAAA	108	XM_040663107.1
TTPA R	GCGAAGACTGGGTGGAAGAA		
BCO1 F	ACAAAGAAGAGCATCCAGAGC	142	NM_001364902.1
BCO1 R	GCCAAGCCATCAAACCAGTG		
CYP26A1 F	ATGGAGCACACACAGGGTAAT	423	NM_001001129.1
CYP26A1 R	GTTGAACTCGTCCTTGTCGGT		
RBP4A F	CTTCAATAACTGGGATGTCTG	287	NM_205238.1
RBP4A R	GGTCTATCTGCCTTTGTCTAAC		

GAPDH: glyceraldehyde 3-phosphate dehydrogenase, TTPA: alpha-tocopherol transfer protein, BCO1: β-carotene oxygenase 1, CYP26A1: cytochrome P450 family 26 subfamily A member 1, RBP4A: retinol-binding protein 4A, F: forward and R: reversed.

**Table 3 animals-12-03156-t003:** Production performance in laying hens fed different oils.

Treatment	CPO	RPO	RBD	PKO	SBO	SEM	*p*-Value
HDEP (%)							
Week 22–25	88.69	87.03	91.67	90.12	87.26	1.49	0.866
Week 26–29	92.74	90.60	93.21	94.52	88.22	1.57	0.764
Week 30–33	93.69	90.60	93.57	93.93	86.43	1.70	0.609
Week 34–37	91.31	88.10	91.07	92.02	85.24	1.51	0.616
Overall	91.61	89.08	92.38	92.65	86.79	1.49	0.713
Egg weight (g/hen/day)							
Week 22–25	54.01	53.47	55.16	54.23	54.75	0.258	0.271
Week 26–29	57.44	55.67	57.25	56.65	57.36	0.275	0.210
Week 30–33	57.99 ^a,b^	56.44 ^b^	57.46 ^a,b^	57.41 ^a,b^	59.12 ^a^	0.279	0.045
Week 34–37	57.51	56.30	57.38	57.04	57.96	0.261	0.348
Overall	56.74	55.46	56.82	56.33	57.24	0.250	0.213
Egg mass (g/hen/day)							
Week 22–25	48.12 ^b^	46.93 ^b^	51.43 ^a^	49.21 ^a,b^	48.04 ^b^	0.439	0.015
Week 26–29	53.50 ^a,b^	50.74 ^b^	53.77 ^a^	53.87 ^a^	50.83 ^b^	0.450	0.034
Week 30–33	54.60	51.40	54.17	54.24	51.86	0.496	0.119
Week 34–37	52.76	49.80	52.63	52.77	49.82	0.454	0.042
Overall	52.24 ^a,b^	49.72 ^b^	53.02 ^a^	52.52 ^a,b^	50.12 ^b^	0.433	0.045
Feed intake (g/hen/day)							
Week 22–25	102.67	100.67	105.48	103.45	103.26	0.667	0.258
Week 26–29	108.42	106.37	108.36	107.82	106.88	0.375	0.329
Week 30–33	106.64	106.52	108.43	107.26	107.20	0.308	0.316
Week 34–37	103.29	103.04	106.86	104.53	104.40	0.616	0.316
Overall	105.25	104.15	107.28	105.76	105.44	0.468	0.336
Feed conversion ratio							
Week 22–25	2.03	2.06	2.03	2.06	2.05	0.018	0.971
Week 26–29	1.99	2.01	1.98	1.96	1.97	0.016	0.908
Week 30–33	1.98	2.03	2.00	1.99	1.97	0.017	0.867
Week 34–37	2.05	2.11	2.04	2.05	2.01	0.018	0.479
Overall	2.01	2.06	2.01	2.01	2.00	0.014	0.664

CPO: crude palm oil, RPO: red palm oil, RBD: refined palm oil, PKO: palm kernel oil, SBO: soybean oil, HDEP: hen-day egg production, FCR: feed conversion ratio. SEM: standard error of means. ^a,b^ Means with different superscripts in the same rows depict significant differences (*p* < 0.05). Experimental unit, *n* = 6.

**Table 4 animals-12-03156-t004:** Egg quality in laying hens fed different oils.

Treatment	CPO	RPO	RBD	PKO	SBO	SEM	*p*-Value
Albumen height (mm)	5.240 ^a^	4.860 ^b^	4.910 ^b^	4.960 ^a,b^	4.820 ^b^	0.048	0.035
Haugh unit	70.20 ^a^	67.44 ^b^	67.27 ^b^	67.90 ^a,b^	66.38 ^b^	0.407	0.035
Shell thickness (mm)	0.385	0.380	0.382	0.382	0.392	0.002	0.418
Shell weight (g)	5.583	5.517	5.510	5.510	5.569	0.033	0.929
Shell percentage (%)	9.354	9.432	9.276	9.276	9.189	0.048	0.580
Yolk color							
RCF	4.848 ^a^	4.915 ^a^	4.527 ^b^	4.624 ^b^	4.359 ^c^	0.036	<0.001
L* (lightness)	61.65	60.55	61.87	61.36	61.01	0.258	0.534
a* (redness)	12.28 ^b^	14.72 ^a^	11.50 ^b^	12.21 ^b^	11.93 ^b^	0.256	<0.001
b* (yellowness)	62.49 ^d^	67.96 ^a^	63.60 ^c,d^	65.56 ^b^	65.40 ^b,c^	0.437	<0.001

CPO: crude palm oil, RPO: red palm oil, RBD: refined palm oil, PKO: palm kernel oil, SBO: soybean oil, RCF: Roche yolk color fan. ^a,b,c,d^ Means with different superscripts in the same row depict significant differences (*p* < 0.05). Experimental unit, *n* = 6.

**Table 5 animals-12-03156-t005:** Serum biochemical profiles in laying hens fed different oils.

Treatment	CPO	RPO	RBD	PKO	SBO	SEM	*p*-Value
Liver and kidney functions							
ALP (U/L)	339.6 ^c^	504.1 ^b,c^	443.5 ^c^	736.3 ^a^	670.0 ^a,b^	44.83	0.004
ALT (U/L)	16.21	20.74	16.45	18.59	19.06	1.835	0.952
AST (U/L)	328.1	363.1	314.8	341.9	295.9	13.99	0.676
GGT (U/L)	25.55	18.98	27.11	21.01	27.24	1.313	0.140
Serum proteins							
TP (g/L)	45.93	45.63	47.73	44.37	48.27	0.784	0.565
Albumin (g/L)	21.44	22.76	22.73	19.71	22.86	0.480	0.166
Globulin (g/L)	24.50	22.87	25.01	24.66	25.41	0.574	0.750
A:G	0.880	1.001	0.913	0.803	0.905	0.029	0.255
Minerals							
Ca (mmol/L)	6.000	5.967	5.730	5.883	6.410	0.127	0.592
P (mmol/L)	1.380	1.867	1.177	1.337	1.430	0.097	0.232
Ca: P	4.396	3.580	4.909	4.444	4.491	0.222	0.483

CPO: crude palm oil, RPO: red palm oil, RBD: refined palm oil, PKO: palm kernel oil, SBO: soybean oil, SEM: standard error of means. ^a,b,c^ Means with different superscripts in the same rows depict significant differences (*p* < 0.05). Experimental unit, *n* = 6.

**Table 6 animals-12-03156-t006:** Feed, liver, serum and yolk retinol, β-carotene and α-tocopherol profiles in laying hens fed different oils.

Treatment	CPO	RPO	RBD	PKO	SBO	SEM	*p*-Value
β-carotene (μg/mL)							
Feed	28.48 ^a^	29.83 ^a^	3.659 ^b^	3.474 ^b^	4.272 ^b^	3.339	<0.001
Serum	1.238	1.251	0.469	1.103	0.546	0.127	0.095
Liver	12.97 ^a^	11.39 ^a^	4.338 ^b^	6.179 ^b^	5.387 ^b^	0.955	<0.001
Yolk	57.78 ^a^	54.17 ^a^	43.34 ^b^	56.42 ^a^	47.26 ^b^	1.643	0.001
Retinol (mM)							
Serum	13.66	14.24	14.94	15.60	15.58	0.354	0.356
Liver	9.637	9.178	7.856	8.352	7.874	0.509	0.799
Yolk	7.562	6.865	5.929	7.048	6.883	0.206	0.138

CPO: crude palm oil, RPO: red palm oil, RBD: refined palm oil, PKO: palm kernel oil, SBO: soybean oil, SEM: standard error of means. ^a,b^ Means with different superscripts in the same rows depict significant differences (*p* < 0.05). Experimental unit, *n* = 6.

**Table 7 animals-12-03156-t007:** Feed, liver, serum and yolk α-tocopherol profiles in laying hens fed different oils.

Treatment	CPO	RPO	RBD	PKO	SBO	SEM	*p*-Value
α-tocopherol (mM)							
Feed	0.829 ^a^	0.737 ^a,b^	0.673 ^a,b^	0.547 ^b^	0.554 ^b^	0.037	0.033
Serum	0.310	0.330	0.463	0.427	0.324	0.040	0.716
Liver	0.121	0.205	0.152	0.144	0.130	0.013	0.295
Yolk	1.788	1.732	1.712	1.770	1.702	0.074	0.997

CPO: crude palm oil, RPO: red palm oil, RBD: refined palm oil, PKO: palm kernel oil, SBO: soybean oil, SEM: standard error of means. ^a,b^ Means with different superscripts in the same rows depict significant differences (*p* < 0.05). Experimental unit, *n* = 6.

**Table 8 animals-12-03156-t008:** Liver retinol, β-carotene and tocopherol gene expression in laying hens fed different oils.

Treatment	CPO	RPO	RBD	PKO	SBO	SEM	*p*-Value
TTPA	0.984	1.519	1.922	1.508	1.000	0.172	0.412
BCO1	0.232	0.286	0.633	0.652	1.000	0.101	0.074
RBP4A	1.908 ^a^	0.980 ^b^	0.968 ^b^	2.067 ^a^	1.000 ^b^	0.143	0.001
CYP26A1	0.131 ^b^	0.448 ^b^	0.170 ^b^	0.374 ^b^	1.000 ^a^	0.095	0.004

TTPA: alpha-tocopherol transfer protein, BCO1: β-carotene oxygenase 1, RBP4A: retinol-binding protein 4A, CYP26A1: cytochrome P450 family 26 subfamily A member 1, CPO: crude palm oil, RPO: red palm oil, RBD: refined palm oil, PKO: palm kernel oil, SBO: soybean oil. SEM: standard error of means. ^a,b^ Means with different superscripts in the same rows depict significant differences (*p* < 0.05). Experimental unit, *n* = 6.

## Data Availability

Not applicable.

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
