# Peer review of "Influence of Dietary Palm Oils, Palm Kernel Oil and Soybean Oil in Laying Hens on Production Performance, Egg Quality, Serum Biochemicals and Hepatic Expression of Beta-Carotene, Retinol and Alpha-Tocopherol Genes"

_animals, 2022, doi:10.3390/ani12223156_

Round 1

Reviewer 1 Report

The current study assesses how different dietary oils can eventually affect laying hens' performance, and health. The manuscript is well written in general and can contribute to the current scientific knowledge. However, some shortcomings should be addressed in order to improve the overall quality.

Introduction

The introduction is a bit short, Authors should provide more information about how CPO, RPO, RBD, PKO are different.

Material and methods

Section 2.1: authors should give other environmental conditions such as the temperature and humidity conditions during the trial.

The authors mentioned that birds were purchased at 16 weeks old but the feeding trial started when the birds were 22 weeks old. So what diet the birds received from 16 weeks to 22 weeks? Please clarify in the text.

Section 2.6: How did authors determine serum biochemicals?? Using kits? An automated machine with dry slide technology? There is no information on the methodology used, please give more details.

Section 2.9: The current literature advises using 2 housekeeping genes (ie: GAPDH and B-actin) and use their geometric mean as a reference for the calculation of the fold changes. Why did the authors only use GAPDH?

Section 2.10: Before applying one-way ANOVA it is recommended to check assumptions such as homoscedasticity and the normality of the distribution, did the authors perform such tests? if yes it should be mentioned in the text along with the test used ie: Levene’s? Shapiro wilk?

The authors have an overall good design, however, representing their design in a figure would enhance reader understanding. Timing should be particularly well represented in the figure.

Results

Section 3.2: If authors collected eggs fortnightly (every 2 weeks) for the egg quality results why are they not presenting the data accordingly and only have one table? For instance, each parameter can be presented in a line graph in a timely manner with the day of collection on the x-axis. Is there a specific reason for pooling the data in a single table? Please explain

All tables: can authors specify the number of replicates used in the caption? for instance (n = XX). This could be helpful for other researchers trying to perform a meta-analysis.

Table 8: authors should be consistent in the use of compact letter display: please put the (a,b,..) in superscript like the other tables.

Author Response

No.

Comment

Responses

Introduction

1

The introduction is a bit short. Authors should provide more information about how CPO, RPO, RBD, PKO are different.

The information on the difference in the extraction process from palm fruit to produce CPO, RPO, RBD and PKO and the differences in properties between CPO, RPO, RBD and PKO are added to the text. Finally, the summary of properties of oils is summarised in the figure.

Material and methods

2

Section 2.1: authors should give other environmental conditions such as the temperature and humidity conditions during the trial.

The environmental temperature and humidity during the feeding trial are added in the manuscript.

3

The authors mentioned that birds were purchased at 16 weeks old but the feeding trial started when the birds were 22 weeks old. So what diet the birds received from 16 weeks to 22 weeks? Please clarify in the text.

The diets received from the 16th to 21st week adaptation are added to the manuscript.

4

Section 2.6: How did authors determine serum biochemicals?? Using kits? An automated machine with dry slide technology? There is no information on the methodology used, please give more details.

The serum biochemicals were determined using kits. The detail on the determination of serum biochemicals is added to the manuscript.

5

Section 2.9: The current literature advises using 2 housekeeping genes (ie: GAPDH and B-actin) and use their geometric mean as a reference for the calculation of the fold changes. Why did the authors only use GAPDH?

We started with both GAPDH and B-actin genes as a housekeeping genes. However, the combination of both housekeeping genes and B-actin alone gave unsatisfactory, inconsistent and high deviation of the expression. However, GAPDH alone provides a good and consistent expression, and we decided to use only GAPDH for this study.

6

Section 2.10: Before applying one-way ANOVA it is recommended to check assumptions such as homoscedasticity and the normality of the distribution, did the authors perform such tests? if yes it should be mentioned in the text along with the test used ie: Levene’s? Shapiro wilk?

Yes, the data were checked for normality using PROC UNIVARIATE on SAS software and determined based on Shapiro-wilk. Detailed information on the data distribution analysis is added to the text.

7

The authors have an overall good design, however, representing their design in a figure would enhance reader understanding. Timing should be particularly well represented in the figure.

The design is represented in the figure and added to the manuscript.

Results

8

Section 3.2: If authors collected eggs fortnightly (every 2 weeks) for the egg quality results why are they not presenting the data accordingly and only have one table? For instance, each parameter can be presented in a line graph in a timely manner with the day of collection on the x-axis. Is there a specific reason for pooling the data in a single table? Please explain

The reason for pooling the egg quality results in one table was mainly due to no significant increase or decrease in the egg quality parameters over time. In addition, it is not our objective to see the changes in egg quality over time and the difference between treatments was seen better seen in pooling the data.

9

All tables: can authors specify the number of replicates used in the caption? for instance (n = XX). This could be helpful for other researchers trying to perform a meta-analysis.

The experimental unit/sample replicates are included in the description or footnotes of the result table.

10

Table 8: authors should be consistent in the use of compact letter display: please put the (a,b,..) in superscript like the other tables.

The compact letter display on Table 8 was changed to superscript form.

Reviewer 2 Report

Simple Summary: Please be more specific when describing the overall results. It will be easier for the reader.

Line 66-67: Expand on why PKO is less common to use in poultry diets than palm oils.

Lines 68-70: Please include the references that you are referring to.

Review the literature again and be sure to incorporate relevant information into the introduction. Perhaps discuss how the use of oils varies by country, region, etc. in poultry diets.

Lines 139-149: The n number for sampling was low. Please justify why only n=6/treatment was selected. At what age were the samples collected? 38 weeks? It needs to be specified in this section too.  

Table 3: Would it be more appropriate to state the weeks based on the actual age of the birds? (Ex: instead of Week 1-4, include Week 22-25 in the table)

Author Response

No.

Comment

Response

1

Simple Summary: Please be more specific when describing the overall results. It will be easier for the reader.

The overall result in the simple summary is written in such a way because we must comply with the word count limitation to prepare the summary. However, the overall result in the simple summary was revised to be more specific.

2

Line 66-67: Expand on why PKO is less common to use in poultry diets than palm oils.

Additional information on the reason for the lesser use of PKO in poultry diet is added in the text.

3

Lines 68-70: Please include the references that you are referring to.

The sentences are revised, and the references to the mentioned studies are added.

4

Review the literature again and be sure to incorporate relevant information into the introduction. Perhaps discuss how the use of oils varies by country, region, etc. in poultry diets.

The literature in the introduction is updated.

5

Lines 139-149: The n number for sampling was low. Please justify why only n=6/treatment was selected. At what age were the samples collected? 38 weeks? It needs to be specified in this section too.

The sample size chosen was satisfactory to represent the treatment group. The deviation of the data in a group was satisfactory. The age of the birds during sample collection was on 37th week and is added to the text (Section 2.5).

6

Table 3: Would it be more appropriate to state the weeks based on the actual age of the birds? (Ex: instead of Week 1-4, include Week 22-25 in the table).

Yes, it is more appropriate to state the weeks based on actual age. Therefore, the table has been updated to the actual age.

Reviewer 3 Report

Dear authors. The results of the research obtained in your work "Influence of dietary palm oils, palm kernel oil and soybean oil in laying hens on production performance, egg quality, serum biochemicals and hepatic expression of beta-carotene, retinol and alpha-tocopherol genes" are very interesting. The work, however, is written at a very low level. Requires a new study. The very introduction to the work will very briefly and chaotically describe the problem of the research topic. There are few references to scientific literature which is very rich. In line 69-70 you indicate that there has been a lot of research, but you are not referring to it. Very few items were also used in the discussion chapter. In total, the work consists of only 30 items of literature cited. The research methodology is quite well written and, in my opinion, does not require any corrections. However, the chapters results and discussion as well as conclusions must be rewritten. The description of the results is made in a few sentences and is very short. Likewise, the discussion is very short and does not indicate possible causes, and there is very little reference to the available scientific literature. The conclusions should also be changed and corrected.

Author Response

No.

Comment

Response

1

The very introduction to the work will very briefly and chaotically describe the problem of the research topic.

The introduction section is revised to supply more information on the research topic. In addition, the research problem is improved. All previous research on the oil of interest was cited and referred to in lines 93-95.

2

There are few references to scientific literature which is very rich. In line 69-70 you indicate that there has been a lot of research, but you are not referring to it. Very few items were also used in the discussion chapter. In total, the work consists of only 30 items of literature cited.

The lack of literature cited in the manuscript was due to very few previous studies on oils of interest, particularly in laying hens. The discussion includes all available previous findings related to the oil of interest in other poultry species, such as broiler chickens and laying ducks. The finding of previous studies with dietary carotenoids and vitamin E in poultry were also discussed.

3

The description of the results is made in a few sentences and is very short.

The description of the results is short because it focuses on the main findings of the specific table or parameters. The lack of significance of most parameters in a table reduced the description length of results.

4

The research methodology is quite well written and, in my opinion, does not require any corrections.

However, the chapters results and discussion as well as conclusions must be rewritten.

Likewise, the discussion is very short and does not indicate possible causes, and there is very little reference to the available scientific literature.

The discussion covered all previous studies related to dietary oils of interest in laying hens. There are few previous studies related to the oils used in this study. Other studies were not comparable for discussion in this current manuscript, such as the blend of different sources of oils and adding the oil with other additives such as vitamins, amino acids, fatty acids or other feed additives.

5

The conclusions should also be changed and corrected.

The conclusion is revised.

Round 2

Reviewer 3 Report

Dear Authors,

the current version of the work is much better. the only caveat I have to work are applications. I suggest connecting sentences from lines 536-537 and 542 with each other. Moreover, in the conclusions, he suggests mentioning the positive effect of RBD on Egg mass (g / hen / day)
